Figure 9: **Parity problem instances: (left)** A sample 8-bit parity problem instance, along with the scratchpad targets. The scratchpad represent the intermediate parity state as the sequence is processed left to right. **(right)** A parity problem instance with the padded scratchpad strategy. Both the input and the scratchpad targets are padded left and right with the same number of padding tokens, such that the relevant bit to attend to while constructing the scratchpad bits is always equidistant even when there are different number of bits in the input.

# A  Data Generation Details

We describe the data generation procedures for the parity and variable assignment tasks in detail.

## A.1  Parity Datasets:

**Synthetic Parity Dataset:** A 8-bit example of the synthetic parity example, along with the corresponding scratchpad targets, can be seen in Figure 9. We added the prefix ">>>" to signify the start of the parity sequence, and the suffix "==" to signify the start of the target or scratchpad tokens. There's no special meaning associated with the particular prefixes and suffixed used.

We experimented with two version of the synthetic parity dataset: In one split, we varied the *number of bits* in the input, and in the other one, we varied the *number of ones*.

- **Varied number of bit split:** To generate the samples in this split, we first sampled the number of bits, then sampled each bit individually from a uniform Bernoulli distribution. For training, we used lengths between 3 and 20, and for validation/testing, we used lengths between 3 and 40.

- **Varied number of ones split:** Here, we fixed the number of bits at 30. To sample each instance, we first uniformly sampled the number of ones, then randomly placed each one in the fixed-length bitstring by randomly shuffling the bits. We used 10 to 20 ones in the training split, and 1 to 30 ones in the validation/test splits.

  **Padded scratchpad:** The padded scratchpad format can be seen in 9. Both the input and the scratchpad targets are padded left and right with the same number of padding tokens respectively, such that the relevant bit to attend to while constructing the scratchpad bits is always equidistant. Moreover the total number of characters/tokens is also kept constant. The number of tokens to pad on the left and right is determined (uniformly) randomly.

The parity datasets contain 1000000 samples.

**Natural Language Parity Dataset:** In order to tap into the natural language understanding capabilities of pretrained language models, we situated the parity task as a "coin flip problem". In this framing, flipping a coin corresponds to 1 and not flipping a coin corresponds to 0. To make the inputs as close as possible to English without occupying too many tokens, we used the sentence templates "Then <NAME> flips." and "Then <NAME> doesn't flip." to represent whether the coin was flipped or not respectively, where "<NAME>" refers to a randomly sampled given name. We also prepended each step with an integer id that count backwards from the total number of steps there are in the input sequence. We've experimented with versions where the integer ids are incremented. This didn't lead to a significant difference in the overall performance.

Two representative sample input-target pairs (including the exemplars) are provided in Figure 10.

## A.2  Boolean Variable Assignment Dataset:

An instance of the variable assignment dataset can be seen in Figure 11. The data generation procedure is aimed at synthesizing large number of qualitatively different programs: (1) A subset of boolean variable assignment operations is uniformly sampled from a large pool of operations. (2) The number

**Input:**
Question The coin is heads up. (4) Then Williams flips. (3) Then Ward flips. (2) Then Valentine doesn't flip. (1) Then Son doesn't flip. Is the coin still heads up?\nSolution Coin is initially heads up. (4) After Williams flips, coin turns to tails. (3) After Ward flips, coin becomes heads. (2) Valentine doesn't flip, so coin stays heads. (1) Son doesn't flip, so coin remains heads. DONE
#
Question The coin is heads up. (4) Then Shade flips. (3) Then Kong doesn't flip. (2) Then Kodi flips. (1) Then Charleston flips. Is the coin still heads up?\nSolution Coin is initially heads up. (4) After Shade flips, coin turns to tails. (3) Kong doesn't flip, so coin stays tails. (2) After Kodi flips, coin becomes heads. (1) After Charleston flips, coin turns to tails. DONE
#
Question The coin is heads up. (4) Then Ka doesn't flip. (3) Then Justice flips. (2) Then Johan flips. (1) Then Jamaica flips. Is the coin still heads up?\nSolution Coin is initially heads up. (4) Ka doesn't flip, so coin remains heads. (3) After Justice flips, coin becomes tails. (2) After Johan flips, coin turns to heads. (1) After Jamaica flips, coin becomes tails. DONE
#
Question The coin is heads up. (4) Then Cy flips. (3) Then Booker flips. (2) Then Ace doesn't flip. (1) Then Ren flips. Is the coin still heads up?\nSolution Coin is initially heads up. (4) After Cy flips, coin turns to tails. (3) After Booker flips, coin becomes heads. (2) Ace doesn't flip, so coin stays heads. (1) After Ren flips, coin turns to tails. DONE
#
Question The coin is heads up. (6) Then Tristan flips. (5) Then Hillary flips. (4) Then Olivia flips. (3) Then Rosa doesn't flip. (2) Then Kurt flips. (1) Then Glenn doesn't flip. Is the coin still heads up?"

**Scratchpad targets:**
"Solution Coin is initially heads up. (6) After Tristan flips, coin becomes tails. (5) After Hillary flips, coin turns to heads. (4) After Olivia flips, coin becomes tails. (3) Rosa doesn't flip, so coin remains tails. (2) After Kurt flips, coin turns to heads. (1) Glenn doesn't flip, so coin stays heads. DONE

**Input:**
Question The coin is heads up. (4) Then Katarina doesn't flip. (3) Then January flips. (2) Then Duke flips. (1) Then Cal doesn't flip. Is the coin still heads up?\nSolution Coin is initially heads up. (4) Katarina doesn't flip, so coin remains heads. (3) After January flips, coin becomes tails. (2) After Duke flips, coin turns to heads. (1) Cal doesn't flip, so coin stays heads. DONE
#
Question The coin is heads up. (4) Then Berry flips. (3) Then Abbi flips. (2) Then Tam flips. (1) Then Ikea doesn't flip. Is the coin still heads up?\nSolution Coin is initially heads up. (4) After Berry flips, coin becomes tails. (3) After Abbi flips, coin turns to heads. (2) After Tam flips, coin becomes tails. (1) Ikea doesn't flip, so coin remains tails. DONE
#
Question The coin is heads up. (4) Then Cain doesn't flip. (3) Then Woody flips. (2) Then Von doesn't flip. (1) Then Thu doesn't flip. Is the coin still heads up?\nSolution Coin is initially heads up. (4) Cain doesn't flip, so coin stays heads. (3) After Woody flips, coin turns to tails. (2) Von doesn't flip, so coin remains tails. (1) Thu doesn't flip, so coin stays tails. DONE
#
Question The coin is heads up. (4) Then Russ flips. (3) Then Williams flips. (2) Then Ward doesn't flip. (1) Then Valentine flips. Is the coin still heads up?\nSolution Coin is initially heads up. (4) After Russ flips, coin becomes tails. (3) After Williams flips, coin turns to heads. (2) Ward doesn't flip, so coin remains heads. (1) After Valentine flips, coin becomes tails. DONE
#
Question The coin is heads up. (8) Then Melissa flips. (7) Then Kevin doesn't flip. (6) Then Steven flips. (5) Then Thomas flips. (4) Then Timothy doesn't flip. (3) Then Kyle doesn't flip. (2) Then Rachel doesn't flip. (1) Then Laura doesn't flip. Is the coin still heads up?

**Scratchpad targets:**
Solution Coin is initially heads up. (8) After Melissa flips, coin turns to tails. (7) Kevin doesn't flip, so coin stays tails. (6) After Steven flips, coin becomes heads. (5) After Thomas flips, coin turns to tails. (4) Timothy doesn't flip, so coin remains tails. (3) Kyle doesn't flip, so coin stays tails. (2) Rachel doesn't flip, so coin remains tails. (1) Laura doesn't flip, so coin stays tails. DONE

Figure 10: **Natural Language parity problem instances:** The coin flip task - which consists of tracking the state of a coin as it undergoes a number of flip or no-flip operations - shares the same underlying problem structure as the parity task.

of operations and number of variables (along with their names, which are single-letter characters) are uniformly sampled based on pre-set hyperparameters. (3) One by one, operations and the variables included in the operations are sampled, while making sure that each added operations retains the semantic correctness of the program.

We now outline the hyperparameters used to generate the *chain-like* and *diverse* splits.

**Chain-like split:**

- **Boolean operators:** assign to *and* with another variable, assign to *or* with another variable, assign to *xor* with another variable, negate

- **Minimum/maximum number of operations:** 3, 19

- **Minimum/maximum number of variables in the program:** 2, 3

**Diverse split:**

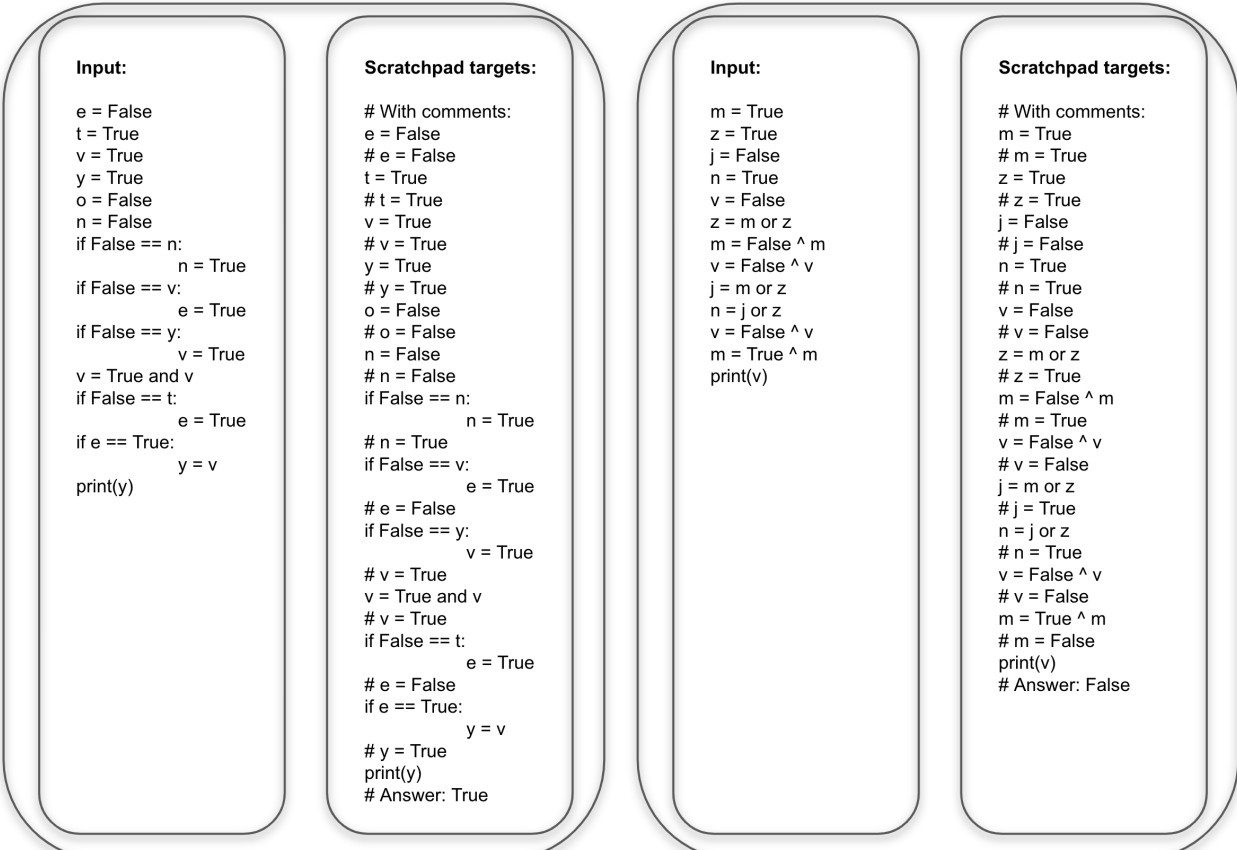

Figure 11: **Variable assignment problem instance:** A problem instance from the Boolean Variable Assignment dataset. The scratchpad strategy consists of outputting the value of the recently updated variable in form of comments. Note that both the input and the scratchpad are valid Python programs.

- **Boolean operators:** assign to *and* with another variable, assign to *or* with another variable, assign to *xor* with another variable, negate, assign to *and* with boolean, assign to *or* with boolean, assign to *xor* with a boolean, conditional assign to a boolean, assign to another variable, conditional assign to another variable
- **Minimum/maximum number of operations:** 8, 32
- **Minimum/maximum number of variables in the program:** 4, 10

The scratchpad format (seen in Figure 11) consists of copying over the input program with comments after each line specifying the value of the recently updated variable. This ensures that the scratchpad itself is a valid Python program and can be used with models pretrained with Python data.

Both splits have 1500000 samples.

## B  Baseline for Vanilla Finetuning on Variable Assignment

Just how weak are the vanilla finetuned models on the OOD lengths on the variable assignment task? We trained baseline models with the same parameter count on a modified version of the variable assignment dataset where the order of the operations were randomly shuffled. While this leaves in some of the spurious features that correlate with the right answer, it completely eliminates the possibility of running a sequential algorithm to get to the final answer.

The results can be found in Figure 12. On OOD data, the performance of the shuffled-ops baseline is on par with the models trained with the clean version of the dataset. Note that the length generalization deficiency that the shuffled ops baselines display is even more dramatic than that of the models'

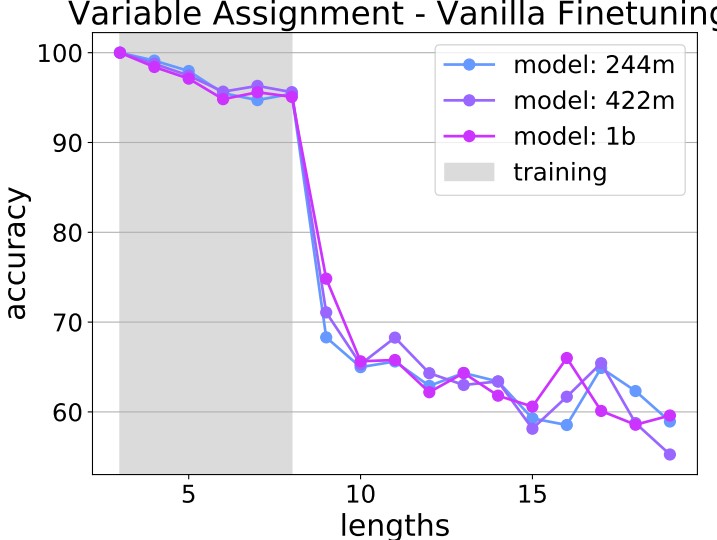

Figure 12: **Results of finetuning on the shuffled-ops variable assignment dataset:** We trained baseline models with the same parameter counts on a modified version of the variable assignment dataset where the order of the operations were randomly shuffled. The generalization gap between in and out-of-distribution data persists here as well, due to transformers' tendency to prefer parallel strategies that don't generalize to larger lengths.

trained with clean data. This is not surprising, as the primary source of lack of length generalization is the transformers' tendency to prefer parallel strategies that don't generalize to larger lengths over picking up sequential algorithm

## C  Experimental Conditions

shuff We outline the training conditions and hyperparameter used in the main experiments.

In our experiments on different datasets, we first did a learning rate sweep over different model sizes, and preferred the largest learning rates that ensured training stability. This is due to the fact that we observed the best length generalization when we used larger learning rates (see Figure 5). We used the AdaFactor optimizer in all of our finetuning experiments [11]. We didn't use dropout [33] in the parity experiments and used a dropout rate of 0.05 in the variable assignment experiments. Our initial experiments suggest that one can often reach similar in and out-of-distribution performance either by training the weights from scratch, or finetuning from the pretrained weights. To keep the experiments consistent, we always initialized training using the pretrained weights. In variable assignment, we did a learning rate sweep over 0.0033, 0.00033 and 0.000033. For parity, we search over learning rate velus of 0.002, 0.0002 and 0.00002. We used a constant learning rate profile all throughout learning. Due to memory constraints, we used a batch size of 32 when training the $64b$ and $128b$ models.

We used greedy decoding in all of our experiments (including few-shot scratchpad ones). We experimented with temperature sampling with reranking based on sentence likelihoods, but found that doing this doesn't lead to qualitatively different results.

## D  Computational Graph Depth is the Relevant Notion of Difficulty on Variable Assignment

*Computational graph depth* captures a more relevant notion of difficulty on the variable assignment task for transformers. In Figure 13, we show how the accuracy of a transformer model evolves on samples of problem instances with different computational graph depths (left), and how the same

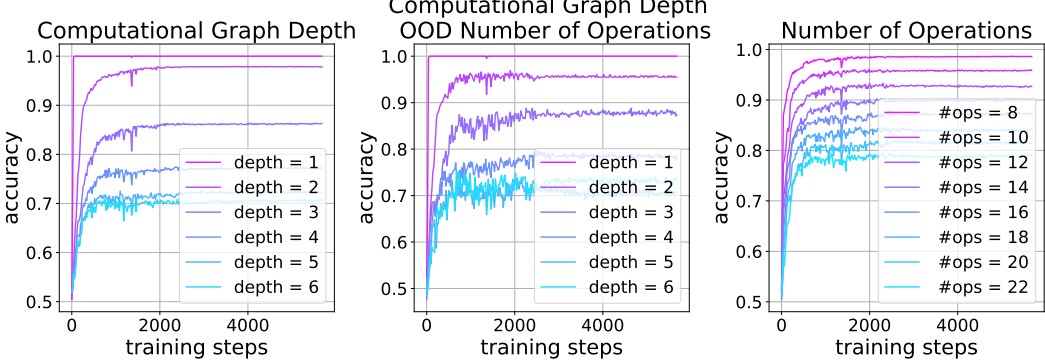

Figure 13: **Evolution of Performance over Training Iterations on Different Computational Graph Depths:** Computational graph depth corresponds to the length of the longest dependency chain linking to the queried variable in the variable assignment task. This quantity captures a more suitable notion of length/difficulty for transformer models. On the **left** plot, we show how the accuracy of a transformer model evolves for problem instances with different computational graph depths. In the *middle*, we show the same, except that we constrain the number of operations in the program to an out-of-distribution number. The accuracy values roughly remain unchanged, indicating that it's not the number of operations, but computational graph depth that determines the difficulty of a problems instance. On the right, we show the evolution of accuracy on instances with different number of operations for reference.

quantity behaves if we fix the number of operations in a program at an out-of-distribution length (middle) [4]. Two takeaways from this analysis are:

- The fact that the accuracy values on samples with different computational graph depths roughly remain unchanged on OOD program lengths indicates that it's not the number of operations, but computational graph depth that captures a more relevant notion of difficulty.

- Transformers initially pick up how to handle programs with a small computational graph depth throughout training and then move to more difficult programs.

# E    Effect of Prompt Style on Few-Shot Finetuning Performance

In Section 5.1, we hypothesized that few-shot finetuning only leads to significant improvements in length generalization performance if the non-finetuned performance on the same task already at a nontrivial level. To provide a sanity check for this, we ran few-shot finetuning using an alternative prompt style for the coin-flip task that yields poor non-finetuned performance. As can be seen in Figure 14, the few-shot finetuned performance shows significant length generalization pathologies. We leave a systematic study of how prompt style affects length generalization as future work.

# F    Distractor Analysis for Scratchpad Strategies

Our analysis in Section 4 indicates that length generalization pathologies persist even when we use the padded scratchpad strategy that makes sure that it's not untrained position encodings and/or the EOS token prediction that causes the aforementioned pathologies. This points to the fact that the transformer doesn't learn to attend to the "right" section of the input and scratchpad that implements the sequential strategy that generalizes to longer lengths — it's thrown off by distractor tokens in the input and/or the preceding scratchpad targets. The distractor tokens at which section of the transformer context window (input or scratchpad) contribute more to the performance deterioration? If we remove all the distractor tokens, can we achieve perfect length generalization?

To answer these questions, we trained four transformer models under the following conditions: (1) We used the padded scratchpad strategy described in Section 4 with no modification, (2) We manually masked the preceding scratchpad tokens that don't contribute to the correct sequential algorithm (i.e.

---

[4]The longest in-distribution number of operations is 15, and we fix this quantity at 20 in the middle plot.

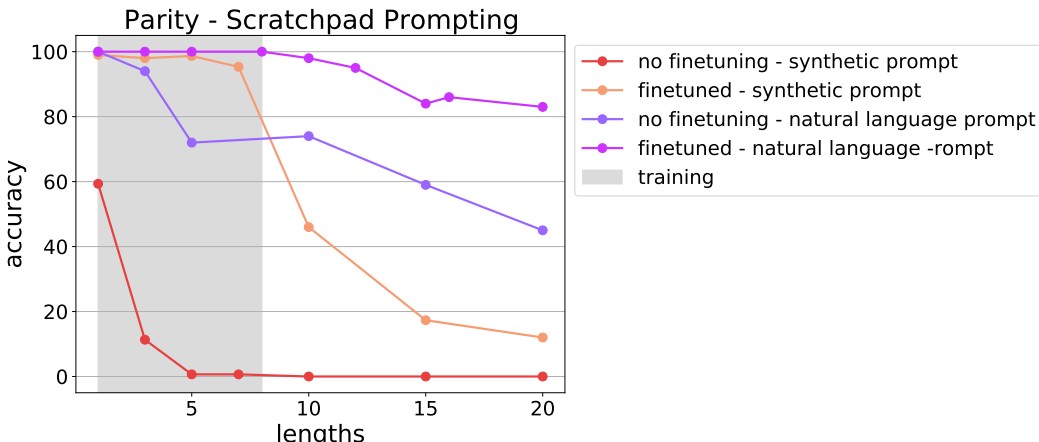

Figure 14: **Effect of prompt style on few-shot finetuning performance:** we evaluated the (few-shot) fine-tuned performance of the pretrained model on an alternative, synthetic prompt style that yields poor performance without any pretraining. We observed that the few-shot finetuned model on this task also shows significant length generalization pathologies. This is in line with our hypothesis that the non-finetuned performance of the base model should be non-trivial for few-shot finetuning to consistently yield strong length generalization results.

we masked the distractor tokens in the input), (3) We masked the input tokens that don't contribute to the correct sequential algorithm, (4) we masked the distractor tokens in both the input and the preceding distractor tokens. To give an example, let's say the input bitstring is "[1 1 0 1 1]", and the model has emitted the scratchpad tokens "[1 0 0]" so far. Masking the distracting sratchpad tokens simply means replacing all but the last scratchpad token with dummy padding tokens: "[x x 0]". Similarly, removing the distracting input tokens corresponds to masking out the part of the input that the network doesn't need to attend to while predicting the next bit: "[x x x 1 x]". Masking both corresponds to removing the distractor tokens in both the input and the target, so that the input and the preceding scratchpad become "[x x x 1 x]" and "[x x 0]" respectively.

The results can be seen in Figure 15. For all four experimental conditions, we plotted the accuracy of the trained models on predicting the scratchpad tokens at different steps for inputs of varying (in and OOD) lengths. We conclude from this experiment that:

- Removing all distractor tokens does result in perfect length generalization.
- The distracting input tokens are hurt length generalization performance more.

Based on this analysis, we conclude that innovations in transformer architectures and/or training methodology/objective that alleviate the issues caused by distractor tokens have a chance at significantly improving length generalization.

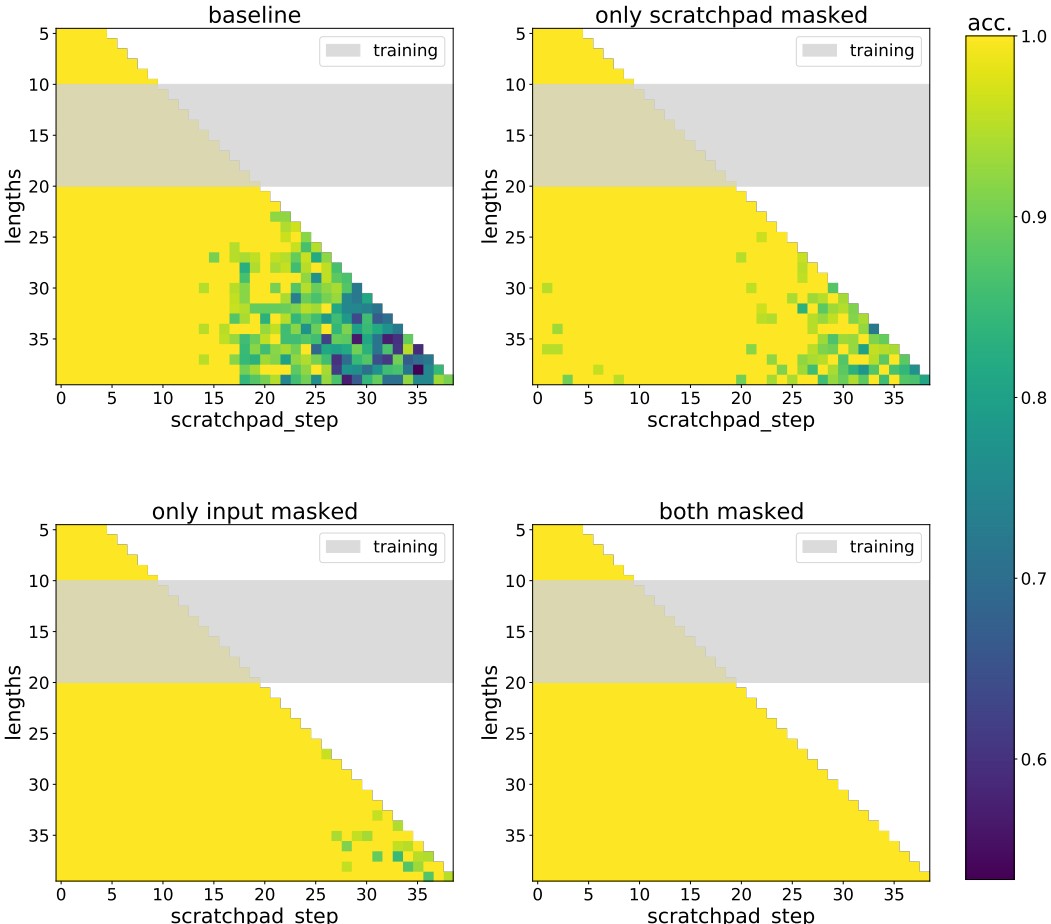

Figure 15: **Distractor analysis:** We trained scratchpad-augmented transformers to solve the parity task, where we systematically masked out the tokens in the input (left bottom) and the scratchpad (right top) that don't need to be attended to while implementing the correct sequential algorithm that solves parity. The plots illustrate the accuracy of the trained models on predicting the scratchpad tokens at different steps for inputs of varying (in and OOD) lengths. This analysis shows that (1) removing all distracting tokens leads to perfect length generalization (right bottom), (2) the distractor tokens in the input contribute more to the length generalization pathologies (right top versus left bottom).