# OpenReview forum: "Exploring Length Generalization in Large Language Models"
_NeurIPS.cc/2022/Conference — NeurIPS 2022 Accept_

### Official Review · Reviewer_jC5u · 2022-07-03

**Rating:** 7
**Confidence:** 4
**Soundness:** 3 good
**Presentation:** 3 good
**Contribution:** 3 good

**Summary:**

This paper is about the performance of large language models (LLMs) when tested against longer sequences than seen in their training data, referred to as length generalization. The authors perform an analysis using two synthetic datasets of their design: 1) parity task (predicting whether the number of 1s in a binary sequence is odd or even), and 2) variable assignment task (predicting the value of a specified variable from a short python snippet). The experiments explore three specific settings with LLMs: 1) fine-tuning, 2) few-shot prompting, 3) scratchpads (which involves augmenting input and predictions with intermediate values). These settings are combined, i.e. fine-tuning + prompting. The main takeaway is that finetuning and prompting alone both fail on length generalization, and that scratchpad prompting (with and without finetuning) is effective for length generalization. At face value, these findings have substantial value for the research community, as scratchpad prompting is a relatively new technique, and is simple and highly effective. That being said, although synthetic tasks are often used in AI, the tasks presented by the authors are not well justified in the text and potentially faulty. Also, while the paper is mostly straightforward to follow, several experimental details are missing and related work is missing.


**Questions:**

Have the authors considered the interplay between finetuning and catastrophic forgetting? Prompt tuning (Lester et al.) has previously been shown to be effective as an alternative to full model finetuning, and I imagine it would have some benefits over few-shot prompting without pitfalls of finetuning. Lester et al. 2021 The Power of Scale for Parameter-Efficient Prompt Tuning

The parity task simply measures counting behavior. This is discussed in detail in McCoy et al. and Suzgun et al., and I believe should apply here as well. The observations in that paper are tied to specific architecture decisions (i.e. non-linearity). How do these prior works relate to the analysis you have presented? McCoy et al. 2020 Does Syntax Need to Grow on Trees? Sources of Hierarchical Inductive Bias in Sequence-to-Sequence Networks. Suzgun et al. 2019, LSTM Networks Can Perform Dynamic Counting.

“...transformers are biased toward learning non-sequential “shortcut” solutions that fail at longer problem instances…” There is a large body of work about the bias of transformers. For instance, see He He et al. 2019. Unlearn dataset bias in natural language inference by fitting the residual; or McCoy et al. 2019. Right for the Wrong Reasons: Diagnosing Syntactic Heuristics in Natural Language Inference. Do these prior works present solutions to what you describe as a shortcut, and if so, why would scratchpad be preferable?

Did you consider alternative synthetic tasks? For instance ListOps (Nangia and Bowman) or the various tasks from Zaremba and Sutskever 2015. Some of these other tasks include interesting properties not related to length, which I suppose could be a distraction from the analysis you set out to perform. Although this perspective potentially contradicts your statement, “This framing [related to Markov Process] applies to a wide range of sequence problems, if not all of them…”


**Limitations:**

This is analysis work and is directly about measuring limitations of various techniques. Perhaps the claims are over stating that we can extrapolate from small synthetic tasks to real world data, but certainly the results provide some value to that end.

I agree with the authors, and don't see negative societal impact caused by this work.

**Strengths And Weaknesses:**

Strengths
+ Controlled experiments to test for length generalization.
+ Detailed experiments in various settings.
+ Analysis demonstrating benefits of existing technique scratchpad, also called chain-of-thought, compared with finetuning and few-shot prompting.

Weaknesses
- Although it is great to use synthetic tasks to measure for model ability, the parity task is not particularly well justified. In the parity task, the history is not necessary if the parity of the history is known. I realize this is by design to mimic a Markov process, but it remains unclear why it is important to think of data this way. Also, the 0% accuracy is concerning and seems easily fixable by constrained decoding like you suggest in footnote 2, perhaps by using a special indicator before predicting the parity label, or even running the sequence twice with continuation of {0, 1} and comparing their probabilities.
- (minor weakness) The experiments explore only a highly specific mode of length generalization related to comprehending long sequences, and not for generating long sequences. Ironically, the best performing setting is the one that requires a variable amount of computation.
- There are specific directions the paper explores that have been explored extensively in prior art: synthetic tasks, bias in data leading to shortcut strategies, and counting behavior in neural nets.
- Some experimental details are missing, like dataset sizes, number of epochs finetuned, and whether the finetuning was done using only the labels or the full sequence.

---

> ### Author Response · Authors · 2022-08-02
> **Response to Reviewer jC5u (3/3)**
>
> > *The parity task simply measures counting behavior. This is discussed in detail in McCoy et al. and Suzgun et al., and I believe should apply here as well. The observations in that paper are tied to specific architecture decisions (i.e. non-linearity). How do these prior works relate to the analysis you have presented? McCoy et al. 2020 Does Syntax Need to Grow on Trees? Sources of Hierarchical Inductive Bias in Sequence-to-Sequence Networks. Suzgun et al. 2019, LSTM Networks Can Perform Dynamic Counting.*
>
> Thank you for pointing us to these very interesting papers! If we understand correctly, the counting behaviour discussed in McCoy et al. is related to recurrent model’s ability to maintain possibly multiple “counters” as it processes tokens left to right. This behaviour is readily available without squashing functions (i.e. randomly initialized LSTMs should already be doing a naive version of this) and might be useful for finding the right algorithm to solve the parity task. **Our most relevant finding is that transformer models also count, but not left-to-right: they count the number of ones in the bitstring bottom-up in a single attention layer, and memorize the parity of the sum.** Taken together, perhaps these findings point to a synergy between recurrence and attention that can capture different inductive biases towards counting. **We added this discussion in the paper.**
>
> We’d also like to mention that both parity and boolean variable assignment tasks admit counting in mod2, which is perhaps qualitatively a bit different than counting integers.
>
> > *“...transformers are biased toward learning non-sequential “shortcut” solutions that fail at longer problem instances…” There is a large body of work about the bias of transformers. For instance, see He He et al. 2019. Unlearn dataset bias in natural language inference by fitting the residual; or McCoy et al. 2019. Right for the Wrong Reasons: Diagnosing Syntactic Heuristics in Natural Language Inference. Do these prior works present solutions to what you describe as a shortcut, and if so, why would scratchpad be preferable?*
>
> Thanks again for the very relevant pointers!
>
> Both the Heuristic Analysis for NLI Systems introduced by McCoy et. al. to capture potentially erroneous entailment relationships, as well as He He et al.’s DRIFT approach for debiasing are very interesting! At a first glance, we don’t identify a method in either of these works that could address length generalization. McCoy’s work is on evaluating systematic biases through which language models (including transformer based ones) make mistakes on natural language inference tasks, and doesn’t propose a fix to these issues. He He et al.’s DRIFT seems to require fitting an initial model on “insufficient features” (such as only the hypothesis sentence in an NLI task), then fits a complementary model that fits the residuals of the initial model on the full set of features. We aren’t quite sure what’d be a good way of picking the insufficient features for the tasks we consider. Please let us know if you have ideas in this regard.
>
> **Few-shot scratchpad provides a very different approach to developing model capabilities on tasks where normally architectural modifications were explored by the research community.** We find this to be a significant finding: that one can go so far without touching the architecture.

---

> > ### Comment · Reviewer_jC5u · 2022-08-08
> > **no subject**
> >
> > The response is very detailed and helpful. Particularly the inclusion of missing training details, further clarifications about experimental design decisions, and discussion of results.
> >
> > To me the fact that the prompt does not always lead to {0, 1} output space suggests that the model has a lot of flexibility for "interpreting" the input sequence, but based on my improved understanding of the work, I see this more so as potential for future improvements more so than a weakness in the current approach.
> >
> > I think this work has great value and should update beliefs for how researchers may want to approach problems about length generalization. For that reason, it is important that readers can understand how the experiments were conducted, especially since model details are scarce besides model size (although perhaps these will be revealed in the non-anonymous revision). The response sufficiently addresses my initial concerns along these lines so I have improved my rating.

---

> ### Author Response · Authors · 2022-08-02
> **Response to Reviewer jC5u (2/3)**
>
> > *(minor weakness) The experiments explore only a highly specific mode of length generalization related to comprehending long sequences, and not for generating long sequences. Ironically, the best performing setting is the one that requires a variable amount of computation.*
>
> This is a very fitting comment. Focusing directly on “comprehending long sequences” as opposed to “generating long sequences” is a decision we made early in our investigation, in order to maintain focus and clarity in our experiments. That being said, **our investigation has inevitably required us to analyze what happens when the models have to output longer scratchpad generations**. In Appendix f (of the updated submission) we **study distractor tokens and their effect on length generalization**. We also ran controlled experiments **to see if fixing the effect of position embeddings and EOS token prediction artifacts solves length generalization** in sequence prediction, and were surprised to observe how little of an improvement these modifications lead to.
>
> > *There are specific directions the paper explores that have been explored extensively in prior art: synthetic tasks, bias in data leading to shortcut strategies, and counting behavior in neural nets.*
>
> Thank you for the very interesting papers you’ve shared later in your review. **We’ve added a section that discusses these works.** See more on our thoughts about these papers below.
>
> > *Some experimental details are missing, like dataset sizes, number of epochs finetuned, and whether the finetuning was done using only the labels or the full sequence.*
>
> **We’ve updated the paper to include these training details.**
>
> Finetuning was done on only the target labels.
> The parity dataset consisted of 1000000 samples and the networks were trained for 20000 gradient steps.
> The variable assignment dataset consisted of 100000 samples in the diverse split, and 1500000 samples in the chain-like split. The networks were trained for 18000 steps.
>
> **Responses to questions:**
>
> > *Did you consider alternative synthetic tasks? For instance ListOps (Nangia and Bowman) or the various tasks from Zaremba and Sutskever 2015. Some of these other tasks include interesting properties not related to length, which I suppose could be a distraction from the analysis you set out to perform. Although this perspective potentially contradicts your statement, “This framing [related to Markov Process] applies to a wide range of sequence problems, if not all of them…”*
>
> Please see our response at the top of the review regarding the suitability of the tasks we picked for assessing length generalization.
>
> > *Have the authors considered the interplay between finetuning and catastrophic forgetting? Prompt tuning (Lester et al.) has previously been shown to be effective as an alternative to full model finetuning, and I imagine it would have some benefits over few-shot prompting without pitfalls of finetuning. Lester et al. 2021 The Power of Scale for Parameter-Efficient Prompt Tuning.*
>
> We have indeed considered this possibility! We do think prompt tuning could be a useful tool in the context of length generalization. There are, however, a couple of reasons why we didn’t prioritize prompt tuning in our experiments:
> * Despite the innocuous-looking number of parameters that get updated during prompt tuning, **even a single prompt can be tuned to match the performance of full-model finetuning on nontrivial tasks** (see Figure 3 in Lester et. al. (2021)) In addition, sometimes heavy regularization in form of l2 regularization and dropout is needed to prevent overfitting to the training set when one uses prompt tuning. **These all suggest that the behaviour of prompt tuning might be qualitatively more similar to fine-tuning than prompting by hand.**
> * **Initialization plays a very important role in prompt tuning** (see the discussion in Section 7 in Lester et. al.), indicating that the optimization landscape is highly non-convex. In the context of prompt tuning for algorithmic tasks, one needs to be careful while picking prompt initializations.
> * We also note that **Dyer et. al. (2021) has shown that scale goes a very long way towards fixing catastrophic forgetting**. We find in our experiments that even 50b models (**we’ve added new results using 50b models since the original submission**) show practically the same generalization trends as 1000x smaller models.
> * Lastly, we did do a **bit of exploratory automated prompt tuning (in token space)** for the chain-of-thought parity task by varying the randomizable elements in the prompt (i.e. which bitstrings are picked for exemplars etc.). While we did see a spread in performance, no single prompt significantly overperformed the others. We didn’t use the highly tuned prompt in our results in order to avoid overclaiming.

---

> ### Author Response · Authors · 2022-08-02
> **Response to Reviewer jC5u (1/3)**
>
> Thank you for your thoughtful critique, and very interesting ideas/pointers. We have updated the paper and provided responses to your questions. We’ve also added more large scale experiments (please see the general response above). We hope that you would consider increasing your score if our response and paper updates sufficiently address your concerns.
>
> * **Suitability of tasks to assess length generalization:** Besides strongly possessing the Markov structure we argue is very helpful in studying length generalization, parity and variable assignment tasks are also suitable for another reason: **both of them admit multiple algorithmic-flavoured solution strategies, with only one (left-right execution) displaying length generalization**. This lets us pinpoint particular inductive biases of transformer models (which builds on the works you’ve cited) and helps us test which interventions (scratchpad and few-shot learning) fix these issues.
> * **Missing citations:** Thank you for bringing the papers you’ve referred to our attention! We have **cited and discussed them in the paper**. See below for specific responses.
> * **Training details:** Thank you for pointing this out. We’ve added the missing training details in the Appendix. We only finetuned the model on the target tokens, which we believe makes more sense for the tasks we consider in the paper.
>
> **General response**
>
> > *Although it is great to use synthetic tasks to measure for model ability, the parity task is not particularly well justified. In the parity task, the history is not necessary if the parity of the history is known. I realize this is by design to mimic a Markov process, but it remains unclear why it is important to think of data this way.*
>
> **We believe that very many real-world tasks can either be viewed as “state tracking tasks” with a Markovian structure, or subsume this ability.** Examples are very many: solving math problems, program execution, reading novels, following multi-step instructions etc. We explicitly focus on parity and variable assignment in our experiments, as these tasks share the same bare-bones Markovian structure as the more complicated tasks above.
>
> The fact that the **models display severe length generalization deficiencies even on tasks as superficially simple as parity and variable assignment** is indicative that we’ve picked the right level of problem complexity to study length generalization.
> There are very many “non-linear” tasks (tasks that go beyond left to right execution - imagine tasks like executing python programs with function calls or loops) that can be “linearized” using an appropriate scratchpad. Studying length generalization deficiencies on explicitly linear tasks like parity shed light on what might go wrong on these more complicated nonlinear tasks.
>
> > *Also, the 0% accuracy is concerning and seems easily fixable by constrained decoding like you suggest in footnote 2, perhaps by using a special indicator before predicting the parity label, or even running the sequence twice with continuation of {0, 1} and comparing their probabilities.*
>
> Footnote 2 in the original submission is regrettably highly misleading. The dominant reason why we see successive 0% and 100% accuracies is that the model simply outputs 0 or 1 based on whichever appears more frequently in the input. For example, when the “number of ones” in the bitstring is less than 10, the majority of the input bits is 0s, so the model simply predicts 0 as the output. This prediction yields 100% accuracy when there’s an even number of 1s in the sequence, and 0% when there’s an odd number. We do observe the phenomena described in the footnote on severely OOD inputs, but not to the extent the original footnote implies.
> **This issue is now fixed in the paper.**

---

### Official Review · Reviewer_4u87 · 2022-07-09

**Rating:** 7
**Confidence:** 3
**Soundness:** 3 good
**Presentation:** 4 excellent
**Contribution:** 3 good

**Summary:**

The paper studies the problem of length generalization in large language models (LLM) w.r.t fine-tuning and in-context learning approaches. To isolate the superficial artifacts in testing data that LLMs could exploit to make correct prediction without generalization, the paper sets up controlled experiments using two synthetic tasks: parity and variable assignment,  Studying two current common practice of fine-tuning and incontext learning on the controlled tasks, the paper found that (1) 	finetuning fails at length generation regardless of model sizes.  When controlling the fixed length of input bit-string in parity task, but varying the number of bit 1, the paper shows that (2) transformer prefers parallel strategy but sequential one and still fails at generalization to the larger number of bit 1. In contrast to previous work, the paper points out that the in-distribution loss is not an indicator for OOD generalization.
In addition to fine-tuning, the paper also studies the length generalization for scratchpad finetuning and scratchpad prompting. The analysis shows that scratchpad prompting generalizes better than the finetuning and scratchpad finetuning. Combining finetuning and scratchpad prompting only improves the parity task.


**Questions:**

In the footnote 2, the authors said that they didn't constrain the the output of the models to 0s and 1s. I wonder what the picture would look like if the output is constrained to 0 and 1. I can imagine that for downstream application using LLMs, it's possible to allow the user to specify the constrain of the output. For instance, in Machine Translation, constrained decoding is desirable [1].

[1] [Fast Lexically Constrained Decoding with Dynamic Beam Allocation for Neural Machine Translation](https://aclanthology.org/N18-1119/). Matt Post, David Vilar. NAACL 2018



**Limitations:**

The authors adequately addressed the limitations and potential negative societal impact of their work.

**Strengths And Weaknesses:**

### Strengths
- The paper sets up controlled experiments with two synthetic tasks to probe the generaliability over sequence length of LLMs in fine-tuning and prompting setups.
- By controlling the sequence length and the number of bit 1 in parity task and the length of the chain in the variable assignment task, the paper draws meaningful conclusions from the experiments.
- The paper also studies three popular approach for downstream applications of LLMs, namely finetuning, scratchpad finetuning, and prompt based method.
- Overall, I like the analysis presented in the paper.

### Weaknesses
- I think that the length generalization in the study is a particular kind of length generalization: linear generalization. By linear generalization, I mean that the information needed to solve the task can be stored incrementally in a window of context. For example, to solve the parity task, one can just keep track of odd/even at each input position. This is reflected in the format of coin tossing where the states can be tracked effectively. Thus, it provides a strong recency signal to the model to complete the task. I wonder if there is a synthetic task that requires non-linear generalization?

---

> ### Author Response · Authors · 2022-08-02
> **Response to Reviewer 4u87**
>
> Thank you for your thoughtful review! Please take a look at our general response above, which outlines some complementary large scale experiments we added to paper and general improvements. Our response is below.
>
> **General response:**
>
> > *I think that the length generalization in the study is a particular kind of length generalization: linear generalization. By linear generalization, I mean that the information needed to solve the task can be stored incrementally in a window of context. For example, to solve the parity task, one can just keep track of odd/even at each input position. This is reflected in the format of coin tossing where the states can be tracked effectively. Thus, it provides a strong recency signal to the model to complete the task. I wonder if there is a synthetic task that requires non-linear generalization?*
>
> This is a very good question! We carefully considered this possibility in the beginning phases of our investigation. Just to provide a concrete context, let’s imagine we’re dealing with a python code execution task where the programs have loops and function calls. We believe this task has the nonlinear nature you are alluding to.
> * **Vanilla finetuning on nonlinear tasks perform poorly even in-distribution:** Previous work explored this by training O(100b) parameter models on the MBPP dataset on code execution [Austin et. al. 2021]. The performance, despite the model scale, is poor. Inspection of model responses show that the model predictions aren’t indicative on step-by-step execution. With this in mind, we chose to focus our investigation on linear tasks.
> * **Scratchpad “linearizes” nonlinear tasks:** Once the right scratchpad strategy is given or learned, running inference on non-linear tasks strongly resembles running inference on linear tasks: all the model has to do is to keep track of the problem state and update it as requested. Therefore, we expect that studying length generalization of scratchpad (finetuned or few-shot) augmented models only on linear tasks goes a long way and highlights qualitatively similar failure modes we’d observe on nonlinear tasks.
> * **Attending to the right part of the input:** There’s one additional difficulty that nonlinear tasks pose: the part of the input that the model has to attend to might change in nontrivial ways (i.e. not just left to right) as the scratchpad tokens are generated (i.e. during while loops, or when a new function is called during execution). Our experiments with parity and variable assignment show that even without this additional difficulty, the models struggle to learn where in the input to attend to in linear tasks on OOD inputs (see Figure 6 and supplementary material E). Recent results (i.e. Minerva) [Lewkowycz et. al., 2022] indicate that chain-of-thought prompting goes a long way towards addressing this issue. Devising clever datasets/experiments to isolate this difficulty would be a very promising future direction.
>
>
> > *In the footnote 2, the authors said that they didn't constrain the output of the models to 0s and 1s. I wonder what the picture would look like if the output is constrained to 0 and 1. I can imagine that for downstream application using LLMs, it's possible to allow the user to specify the constrain of the output. For instance, in Machine Translation, constrained decoding is desirable [1].*
>
> The footnote before the updated version of the paper is regrettably misleading. The dominant reason why we see successive 0% and 100% accuracies is that the **model simply outputs 0 or 1 based on whichever appears more frequently in the input**. For example, when the “number of ones” in the bitstring is less than 10, the majority of the input bits is 0s, so the model simply predicts 0 as the output. This prediction yields 100% accuracy when there’s an even number of 1s in the sequence, and 0% when there’s an odd number. We do observe the phenomena described in the footnote on severely OOD inputs, but not to the extent the original footnote implies.
> The main reasons for not constraining the output space are: 1) we wanted to remain as task-agnostic as possible. In many cases (i.e. imagine an open-ended dialogue application) we don’t have enough prior information about the inputs to put meaningful limits on the output space, 2) The model putting little probability mass on the only tokens it’s finetuned to output is an interesting failure mode we wouldn’t have been able to identify if we constrained the output space.
>
> We’d also like to add that we’ve added new finetuning (vanilla and scratchpad) experiments that use models of 50b size. This still remain the same, even at this very large scale. Other improvements are listed in the general response.

---

### Official Review · Reviewer_eFNS · 2022-07-10

**Rating:** 7
**Confidence:** 5
**Soundness:** 4 excellent
**Presentation:** 3 good
**Contribution:** 3 good

**Summary:**

Inputs to transformers can be categorized by many different properties. Here the authors note that longer inputs might require more 'reasoning' steps and therefore they evaluate multiple different strategies, through the lens of input hardness (decided by the length).

They analyze chain-of-thought/scratchpad prompting on a few different artificial tasks and show that transformers have a hard time extrapolating to longer input lengths (and sometimes even to shorter ones).

I feel like the contribution is important and the discussions here are new and interesting. The main downside is that the tasks are artificial and pretty far away from anything that could be useful in production (and that's OK).

**Questions:**

There are many ways to 'prompt' for each task and it's hard to draw conclusions based on one prompt. For example, for the parity prompt, I think you may achieve better results if you use a prompt that's more similar to the one you used for the variable assignment problem- as in- compute the intermediate steps within the sequence of bits.

1: parity-1

1: parity- 0

0: parity- 0

......

I'm wondering if you've tried that.



**Limitations:**

The authors adequately addressed the limitations and potential negative societal impact of their work.

**Strengths And Weaknesses:**

Strengths:
1. New discussion of the ability of transformers to extrapolate to longer sequences with scratchpad prompting and/or finetuning.
2. Interesting analysis of why the models that fail do so in some cases.

Weaknesses:
1. Only artificial toy tasks are experimented with. It's not super clear which more 'realistic' tasks can be used here, but python programs might be applicable.

---

> ### Author Response · Authors · 2022-08-02
> **Response to Reviewer eFNS**
>
> Thank you for your supportive opinions and good questions. We’ve improved the paper and added large scale experiments (see general response above). Please see our response below!
>
> **General response:**
>
> > *Only artificial toy tasks are experimented with. It's not super clear which more 'realistic' tasks can be used here, but python programs might be applicable.*
>
> While **we purposefully picked tasks on which we could run careful and deliberate experiments to preserve our clarity of thought**, extending the evaluations to more messy tasks would certainly have value as future work. We did indeed consider experimenting with more sophisticated Python programs. Upon initial investigations and consultation with researchers who’ve studied code generation and execution in depth, we predicted that the failure modes we observe in the variable assignment programs would still remain, and be compounded by other pathologies that’d make our analyses less crisp.
>
> **Response to Questions:**
>
> > *There are many ways to 'prompt' for each task and it's hard to draw conclusions based on one prompt. For example, for the parity prompt, I think you may achieve better results if you use a prompt that's more similar to the one you used for the variable assignment problem- as in- compute the intermediate steps within the sequence of bits.  [...] I'm wondering if you've tried that.*
>
> **We did indeed run exploratory analyses trying almost exactly this**: we evaluated our variable assignment models on the “parity version” of variable assignment task. As you’ve mentioned, using no-op (i.e. x = x) and negation (i.e. x = not x) one can recover the parity task using the variable assignment task as the skeleton. We found that the length generalization pathologies remained roughly identical.
> This doesn’t rule out that there might be prompting strategies that prove more effective than the ones we tried in the paper - in fact, we expect this to be true in the future as our models get bigger and more capable. That being said, we’re confident about the future validity of our main takeaways in the paper, summarized in Table 1.
>
> **Additional experiments:** We also repeated our finetuning experiments on a model of size around 50b, and found that the behaviour of the model is almost identical to those of smaller sizes.
>
> We appreciate that you’ve found our contribution important and the discussions new and interesting. Please let us know if our responses above could alleviate your concerns.

---

### Official Review · Reviewer_zzMA · 2022-07-11

**Rating:** 7
**Confidence:** 4
**Soundness:** 4 excellent
**Presentation:** 3 good
**Contribution:** 4 excellent

**Summary:**

This paper shows that large language models (LLMs) trained on tasks of a certain length fail to generalize to longer sequence lengths. Long sequence length tasks introduce the difficulty of 1) extrapolation and 2) limited number of examples. They show that a combination of fine-tuning, scratchpad, and few-shot can partially overcome this deficiency up to 1b model scales.

I believe this paper has the raw material to be significant, but it requires some updates to the plots and writing. This is an important analytical piece that is part of the solution for useful, long-range models. After the questions and limitations I've raised are sufficiently handled, I am willing to increase my score.


**Questions:**

- Related to the above weaknesses comment, I don't know precisely what you did to make this work? Your best method is described as "Fine-tuning + prompting + scratchpad". Does this mean you constructed examples using the scratchpad, fine-tuned on that data, but then at inference time, you additionally provided a few examples in the context? Please contrast this explicitly with "Fine-tuning + scratchpad" which performs poorly.
- I don't understand the "short-cut" solution explained here: "involves counting the number of ones in the input, and then thresholding the output". What does "thresholding the output" mean? If the model counts the number of 1s and reports the parity, isn't that the true algorithm that generalizes?
- In Figure 5, why were your fine-tuned models ever outputting tokens different than 0 or 1? In my experience, the label distribution is one of the first things learned when fine-tuning a LLM and anything other than that being output usually indicated a bug or an undertrained model. How did you confirm there was no error?
- How are the authors supporting their claim about poor length generalization to the 100B model scale? Please cite or show data. Note that Codex models were only trained up to 12B parameters.


**Limitations:**

None listed.

**Strengths And Weaknesses:**

Strengths
- This addresses an important problem in the deployment and useful of LLMs
- Naively training on more long-sequence lengths tasks is a solution, however, the number of tasks with very long sequences often diminishes.
- This work crucially focuses on the scaling properties of these techniques, rather than focusing on a single model scale.
- They examine the "short-cut" solutions of the models to determine how they can do nearly perfect in the sequence range but rapidly diminish outside of it.

Weaknesses
- Unless I'm misunderstanding, it seems that many figures are computing the accuracy over a __single__ sequence. Can you please construct a full test set and compute the average accuracy for each sequence length (potentially including error bar ranges if it's not too cluttered). Figure 5 was especially confusing seeing accuracy fluctuate from 0% to 100% accuracy.
- Is it possible your models are undertrained? Did you hold fixed the number of examples of the number of training steps? If the latter, please rerun each such that the stopping condition is peak validation accuracy. I'm noticing that for both cases when the learning rate is fairly high, the larger batch size outperformed the smaller batch sizes. If your models turn out to be undertrained, please redo the figures as well as adjust the writing elsewhere in the paper. For instance, the claim "Surprisingly, different hyperparameter choices for finetuning have a large effect on length generalization performance" may be explained by in-distribution examples being learned first followed by out-of-distribution examples and sufficient training normalizes these differences.
- I find the constant switching of "chain of thought" vs. "scratchpad" confusing. For instance, it switches between both terms in Table 1 and its caption. It's good to describe that these are similar approaches, but perhaps then tell the reader that you're sticking with one for ease of reading.
- Figure 6 (right) is very noisy and may not be suitable for a conference publication, as is. Please try to collect statistics and create smoothed plots with error bars.

Suggestions
- In addition to plotting the accuracy, an interesting appendix figure might be to plot the log-probability to the correct token on a log-log axis. This, in addition to reducing noise by including many examples per sequence length, may actually reveal non-trivial scaling properties (I guess it might). If so, please adjust the writing elsewhere in the paper and the Table. You might say it's "poor scaling with model size", but the data might not support definitive claims like "We find that in the finetuning regime, scaling data, model sizes, and compute does not improve length generalization" (L49).
- Reverse the plot order for vanilla fine-tuning and scratchpad fine-tuning.
- For each row of Table 1, please describe with an example and show the empirical results (at least in supplementary material). As a reader, I'm left wondering where this near-perfect approach appears and where I can see the failed results of "Prompting", "Fine-tune + Prompting", etc. Ensure all claims are backed with empirical results. If it's too long, put in supplementary.
- Put Figure 2 in supplementary and move up one of the results plots.
- Provide an example using either the parity or variable assignment task for the "Instance Length as Number of Steps in a Markov Process" section.

---

> ### Author Response · Authors · 2022-08-02
> **Response to Reviewer zzMA (2/2)**
>
> **Suggestions:** Thank you for your constructive suggestions! We’ve already baked in a number of them in our writeup.
>
> **Responses to Questions:**
> > *Related to the above weaknesses comment, I don't know precisely what you did to make this work? Your best method is described as "Fine-tuning + prompting + scratchpad". Does this mean you constructed examples using the scratchpad, fine-tuned on that data, but then at inference time, you additionally provided a few examples in the context? Please contrast this explicitly with "Fine-tuning + scratchpad" which performs poorly.*
>
> “Finetuning + scratchpad” refers to finetuning the model to output the given scratchpad. In other words, the model is given inputs X and is trained to produce scratchpads S and targets T. At test time, the model is expected to output the scratchpad tokens, then output the final prediction.
> * Input: X
> * Finetuning targets: S-T
>
> “Finetuning + prompting + scratchpad” refers to both few-shot prompting the model and in addition finetuning it to output the asked scratchpad tokens. In other words, the input to the model is a few input-scratchpad-target triplets in addition to a new query input, and the model is trained to output the scratchpad and target tokens.
> * Input: X1-S1-T1; X2-S2-T2; X3
> * Finetuning targets: S3-T3
>
> > *I don't understand the "short-cut" solution explained here: "involves counting the number of ones in the input, and then thresholding the output". What does "thresholding the output" mean? If the model counts the number of 1s and reports the parity, isn't that the true algorithm that generalizes?*
>
> **By “thresholding”, we simply mean memorizing.** For in-distribution problem instances, the model simply counts the number of ones in the input (say 15) and memorizes the parity of that count (in this case, “odd”). When the input has an OOD number of ones that the model has not seen before (say 25), it produces erroneous predictions.
>
> > *In Figure 5, why were your fine-tuned models ever outputting tokens different than 0 or 1? In my experience, the label distribution is one of the first things learned when fine-tuning a LLM and anything other than that being output usually indicated a bug or an undertrained model. How did you confirm there was no error?*
>
> We opted to keep the finetuning pipeline as general as possible, therefore didn’t restrict the output of the model to be 0s or 1s. **Note that the finetuning labels were still restricted to only the tokens corresponding to 0 or 1.** Indeed, on the in-distribution training and test distributions, the model always outputs 0 or 1 without any single exception. However, on the most extremely out-of-distribution problem instances, the model behaviour shifts enough to occasionally output other tokens. (Also, the footnote besides Figure 5 is a bit misleading - we’ve fixed it in the revision.)
>
> > *How are the authors supporting their claim about poor length generalization to the 100B model scale? Please cite or show data. Note that Codex models were only trained up to 12B parameters.*
>
> **We’ve added new results on all of our finetuning curves using 53B models.** The trends are almost entirely unchanged.

---

> ### Author Response · Authors · 2022-08-02
> **Response to Reviewer zzMA (1/2)**
>
> Thank you for your encouraging opinions! We have substantially improved the quality of the writing and hopefully addressed your concerns in our response below. Please take a look and see if they amount to an increase in your score!
>
> *Responses to weaknesses:*
>
> > *I believe this paper has the raw material to be significant, but it requires some updates to the plots and writing.*
>
> **We’ve substantially improved the writing and exposition in the new version.** We’ve also improved some of the plots as well as their ordering. Please take a look!
>
> > *Unless I'm misunderstanding, it seems that many figures are computing the accuracy over a single sequence. Can you please construct a full test set and compute the average accuracy for each sequence length (potentially including error bar ranges if it's not too cluttered).*
>
> Figure 5 was especially confusing seeing accuracy fluctuate from 0% to 100% accuracy.
> All of the figures in the paper are actually **generated using test sets of many examples**. We were also initially surprised about the fluctuations between 0 and 100% in Figure 5! The explanation of this phenomenon is the following: when processing OOD bitstrings, the model simply outputs 0 or 1 based on whichever appears more frequently in the input. For example, when the “number of ones” in the bitstring is less than 10 (i.e. OOD), the majority of the input bits is 0s, so the model simply predicts 0 as the output. This prediction yields 100% accuracy when there’s an even number of 1s in the sequence, and 0% when there’s an odd number. The caption in our initial submission was quite misleading - we’ve fixed it in the updated version.
>
> > *Is it possible your models are undertrained? Did you hold fixed the number of examples of the number of training steps? If the latter, please rerun each such that the stopping condition is peak validation accuracy. I'm noticing that for both cases when the learning rate is fairly high, the larger batch size outperformed the smaller batch sizes. If your models turn out to be undertrained, please redo the figures as well as adjust the writing elsewhere in the paper. For instance, the claim "Surprisingly, different hyperparameter choices for finetuning have a large effect on length generalization performance" may be explained by in-distribution examples being learned first followed by out-of-distribution examples and sufficient training normalizes these differences.*
>
> We were careful in **picking training durations such that the in-distribution validation accuracy roughly converges**. In every experiment we ran, the validation accuracy increased monotonically over the training iterations - we never observed “overfitting” (i.e. last checkpoint was always the best).  This could be due to the algorithmic nature of the tasks we considered in the paper. **We didn’t use OOD performance during model selection**, as this constitutes as “peaking at the test conditions”. We also experimented with **training the networks for significantly longer** than it takes to get roughly the best validation accuracy possible, expecting that perhaps OOD performance would change during this process. This didn’t lead to material differences on the OOD performance (i.e. it remained similarly poor).
>
> > *I find the constant switching of "chain of thought" vs. "scratchpad" confusing. For instance, it switches between both terms in Table 1 and its caption. It's good to describe that these are similar approaches, but perhaps then tell the reader that you're sticking with one for ease of reading.*
>
> Thank you for pointing this out. **We’ve edited the paper to ensure consistency.** Whenever warranted, we used both terminologies to avoid any confusion.
>
> > *Figure 6 (right) is very noisy and may not be suitable for a conference publication, as is. Please try to collect statistics and create smoothed plots with error bars.*
>
> **We’ve improved the plot in the original submission.** The main changes were: 1) averaging the results obtained from different instance lengths, 2) putting error bars and 3) considering consecutive three positions when reporting statistics. We believe this version is much less confusing and clearly makes the point that the error rate doesn’t undergo drastic changes as we move to larger positions (unlike regular fine-tuning).

---

### Author Response · Authors · 2022-08-02
**General Response**

We thank all the reviewers for their encouraging reviews! We appreciate the consensus regarding the importance of length generalization as a crucial capability of language models and the positive feedback we got regarding our contributions.  We believe our results (regarding limitations of finetuning and the promise of few-shot scratchpad) have important implications for the NeurIPS community working on language models and reasoning.

We’d like to highlight some improvements we’ve made to the paper:
* **Large-scale experiment:** We ran additional finetuning experiments on a 50b model. Despite its enormous scale, the length generalization trends of this model, whether vanilla finetuned or scratchpad finetuned, are basically identical to that of other models we reported in the initial submission.  We believe this is conclusive evidence that scaling alone doesn’t mitigate length generalization pathologies, at least using model sizes available today.
* **Sanity check experiment:** We also ran a sanity check experiment that supports our hypothesis that few-shot finetuning results are only significantly superior to standard finetuning if the few-shot performance of the model is nontrivial to begin with.
* **Polished writing and plots:** We have improved the writing, the document structure and some plots in response to reviewer comments.

---

### Meta-Review · Area_Chair_Pjod · 2022-08-23

**Recommendation:** Accept
**Confidence:** Certain

**Metareview:**

This paper studies the problem of length generalization for LLMs, and shows that fine-tuning and prompting alone are not sufficient to address it. It introduces a "scratchpad" extension to LLM generation, which prompts the model to produce and condition on intermediate values during reasoning tasks, is a promising way of addressing this issue. The reviewers were unanimous in recommending acceptance, and thus I am happy to follow suit.

**Award:**

No

---

### Decision · Program_Chairs · 2022-09-14

Accept